# Fetal Fibroblast Transplantation via Ablative Fractional Laser Irradiation Reduces Scarring

**DOI:** 10.3390/biomedicines11020347

**Published:** 2023-01-26

**Authors:** Shigeki Sakai, Noriko Aramaki-Hattori, Kazuo Kishi

**Affiliations:** Department of Plastic and Reconstructive Surgery, Keio University School of Medicine, 35 Shinanomachi Shinjiku-ku, Tokyo 160-8582, Japan

**Keywords:** skin, scar, fractional laser, cell transplantation, fibroblast, fetal, sphere

## Abstract

Scar treatments include fractional laser treatment, cell transplantation, surgery, skin needling, and dermal fillers. Fractional laser treatments are used to reduce scarring and blurring. Cell transplantation is promising, with mature fibroblasts and adipose-derived stem cells being used clinically, while embryonic fibroblasts are used experimentally. Herein, we developed a combination of ablative CO_2_ (carbon dioxide) fractional laser and cell transplantation for the treatment of scars. Eight-week-old male C57Bl/6 mice were used to create a full-layer skin defect in the back skin and create scars. The scar was then irradiated using a CO_2_ fractional laser. The cells were then transplanted onto the scar surface and sealed with a film agent. The transplanted cells were GFP-positive murine fetal fibroblasts (FB), fetal fibroblasts with a long-term sphere-forming culture (LS), and fetal skin with a short-term sphere-forming culture (SS). After transplantation, green fluorescent protein (GFP)-positive cells were scattered in the dermal papillary layer and subcutis in all the groups. LS significantly reduced the degree of scarring, which was closest to normal skin. In conclusion, the combination of ablative fractional laser irradiation and fetal fibroblast transplantation allowed us to develop new methods for scar treatment.

## 1. Introduction

Several types of cutaneous scarring have been reported. These include atrophic scars, hypertrophic scars, and keloids, which are characterized primarily by fibrosis, loss of skin texture, lack of appendages, and changes in color tone [1,2]. Surgical procedures to reduce scarring are currently limited to narrowing the scar width to make it less noticeable. Other treatments that effectively improve scarring and blurring include fractional laser therapy and cell therapy. Topical fibroblast transplantation therapy improves skin texture [3], and fetal dermal fibroblasts can regenerate skin [1,4,5]. In addition, we reported that hair can be regenerated by co-transplanting hair follicle cells and sphere-forming fibroblasts [6]. When cells form spheres, they stop proliferating and enter a state called cell hibernation, in which they can survive for long periods. Pluripotency is observed in various stem cells in vivo. Fibroblasts are transformed into a state that is equivalent to mesenchymal stem cells in vitro by forming cell spheres. The authors studied cell spheres during hibernation and found in a preliminary experiment that fibroblast aggregates can differentiate into nerve, fat, bone, and cartilage when induced to differentiate. Sphere formation by different culture methods allows cells to maintain their undifferentiated status and facilitates skin regeneration [7,8]. Therefore, we designed an experiment using fibroblast-sphere transplantations. This study aimed to devise a treatment combining ablative CO_2_ fractional laser treatment with cell transplantation to investigate scar treatment.

## 2. Materials and Methods

### 2.1. Animal Experiment

The study protocol was reviewed and approved by the Keio University Institutional Animal Care and Use Committee of Keio University School of Medicine (approval number:12090–(5)). All experiments were performed following the Institutional Guidelines for Animal Experimentation of Keio University.

Eight-week-old male C57Bl/6 mice were purchased from Sankyo Labo Service Corporation, Inc. The mice were housed in individual cages with room lights on a 12 h cycle. Anesthesia was induced by isoflurane inhalation at a concentration of 5% and maintained at 2–3%. The back skin of the mice was cut with scissors to create a 2 × 2 cm square total skin defect. The wounds formed crusts and epithelialized within 3–4 weeks. Starting from the second month, the scar, which had a clear boundary surrounded by a few hairs, was used for the experiment (Figure 1a). The second-month scar area was irradiated with a CO_2_ fractional laser (SmartXide DEKA Tokyo, Japan). The irradiation conditions were as follows: irradiation mode, DOT; power, 5 W; shape, square; size, 40%; ratio, 10/10; aiming, 60%; smart stack, 1; scan mode, normal; exposure mode, single; emission mode, HP; spacing, 1000 µm; density, 26.7%; fluence, 15.30 J/cm^2^; and pulse energy, 55.1 mJ. Immediately after irradiation, cultured cells were applied to the entire irradiated area and sealed with a film agent (Perme roll^®^ NITTO Tokyo Japan). After 4 weeks, the tissue was collected to check whether the cells had been taken or not. The control group was treated with fractional laser irradiation and protected with film only. They were sacrificed, and the backs of the mice were shaved. The skin was cut with scissors, and a whole skin sample of exactly 2 × 2 cm square was taken.

### 2.2. Fibroblast Culture and Transplantation

Dermal tissue collected from the embryonic day E-17 fetal skin of green fluorescent protein-(GFP)-positive mice purchased from Sankyo Labo Service Corporation, Inc., was cultured using the explant method in 10% bovine serum albumin (BSA), 1% 4-(2-hydroxyethyl)-1-piperazineethanesulfonic acid (HEPES), penicillin/streptomycin, and low glucose Dulbecco’s modified eagle medium (DMEM) in 10 cm dishes. Subsequently, the cells were used after three passages. DMEM/F12 + B27 (Gibco-BRL, Thermo Fisher Scientific, Waltham, MA, USA) 20 ng/mL + epidermal growth factor (EGF) (FUJIFILM Wako pure Chemical, Co., Osakai, Japan) and 40 ng/mL basic fibroblast growth factor (bFGF) (Kaken Pharmaceuticals, Tokyo, Japan) were used to culture cells in the subsequent experiments.

Fibroblasts were administered to the following three groups: the cultured in normal condition fibroblast (FB) group; the cultured with long-term sphere culture (LS) group where the fetal fibroblasts were cultured for 3 weeks in a non-adherent culture dish to create spheres; and the fetal skin with short-term sphere-forming culture (SS) group. The whole skin layer of E-17 was collected, treated with collagenase type 1, and cultured for 3 days in a non-adherent culture to create spheres. For the FB group, 1 × 10^7^ fibroblast cells were administered to each mouse. For the LS group, 1 × 10^7^ fibroblast cells were equally distributed in 96 non-adherent wells, collected three weeks later (Figure 1a), and administered to each mouse. The fetal skin of the SS group was incubated with collagenase type 1 for 45 min at 37 °C, and the blood cells and foreign substances were removed. Then, 1 × 10^7^ cells were evenly distributed in 96 non-adherent wells and collected three days (Figure 1b) later and administered to each mouse. All cells were cultured at 37 °C in a 5% CO_2_ and 20% O_2_ atmosphere.

### 2.3. MMTSS

The Mouse Masson Trichrome Scar Scale (MMTSS) was developed based on the Manchester Scar Scale [9] to assess skin conditions. We calculated the condition of the epidermis and the directionality, density, and maturity of the collagen fibers in the dermis, with a score range of 0–14 (2 + (4 × 3)) (Table 1).

### 2.4. Immunofluorescence Staining and Immunohistochemistry

The collected tissues were fixed with 4% formaldehyde and embedded in paraffin. Paraffin sections of 5 µm thickness were then prepared. Deparaffinization was performed as follows. The slides were washed and deparaffinized three times with xylene for 5 min at room temperature at 20 °C, immersed twice in 100% ethanol, successively in 95%, 70%, and 50% ethanol (3 min each), and finally rehydrated at room temperature at 20 °C.

GFP was used for immunofluorescence staining. All other sections were blocked with protein block (code x0909, Dako Denmark A/S, Glostrup, Denmark) for 15 min, followed by overnight incubation at 4 °C with rabbit anti-GFP antibody (dilution rate of 1:1000, Abcam, Cambridge, UK). After washing with phosphate-buffered saline (PBS) for 5 min, the sections were stained with a secondary anti-rabbit Cy3 antibody (diluted 1:500, Millipore, Billerica, MA, USA). Nuclear staining was performed using 4′, 6-diamidino-2-phenylindole (DAPI) (Thermo Fisher Scientific, Waltham, MA, USA). The samples were visualized using a BX51 microscope with a DP70 camera (OLYMPUS, Co., Tokyo, Japan) and a BZ-X700 fluorescence microscope (KEYENCE, Co., Osaka, Japan).

For the histopathological assay, the sections were stained with Masson’s trichrome (MT) and Elastica van Gieson (EVG). All reagents were purchased from FUJIFILM Wako Pure Chemical Co. (Osaka, Japan). For the MT staining, the sections were first incubated in mordant for 30 min, before rinsing once in distilled water. Equal parts of Weigert’s (A) and Weigert’s (B) were mixed, and the slides were stained with Weigert’s iron hematoxylin for 5 min. The slides were rinsed in running tap water for 5 min, followed by distilled water. The slides were incubated in phosphomolybdic/phosphotungstic acid solution for 50 s to differentiate, then immediately incubated in Orange G for 1 min. An acetic acid solution (1%) was applied to the slide, followed by Masson B for 20 min without rinsing. Again, the slides were briefly incubated in acetic acid solution. The slides were then incubated in 2.5% phosphotungstic acid solution for 10 min, followed by brief incubation in acetic acid solution. Finally, the aniline blue solution was applied to the slide for 3 min and then briefly incubated in acetic acid solution. For EVG staining, Weigert’s resorcinol fuchsin stain solution was added for 30 min. The slides were washed three times in 100% alcohol, followed by running water for 3 min. Equal parts of Weigert’s (A) and Weigert’s (B) were mixed, and the slides stained with Weigert’s iron hematoxylin for 5 min. The slides were washed under running water for 3 min. Van Gieson’s solution was added to the slides for 3 min, before washing with 70% alcohol three times. The samples were dehydrated twice with 95% and 100% ethanol (5 min each).

For immunohistochemical staining, α-smooth muscle actin (α-SMA) was activated with 0.3% hydrogen peroxide (H_2_O_2_) for 30 min. After washing three times with PBS, the sections were incubated with a blocking solution (code x0909, Dako) for 15 min. Primary antibodies were incubated with α-SMA (1:400) (Clone 1A4, Dako Denmark A/S, Glostrup, Denmark) overnight at 4 °C and washed three times for 5 min in PBS. Horse anti-mouse IgG antibody (Vector Laboratories, Newark, CA, USA) was used as the secondary antibody with an avidin-biotinylated peroxidase complex (ABC) kit (Vector Laboratories Inc., Burlingame, CA, USA) at a dilution of 1:200 and incubated at room temperature at 20 °C for 30 min. After washing three times with PBS for 5 min each, the sections were incubated with the Vectastain ABC Kit (Vector Laboratories, Inc., Burlingame, CA, USA). Then, the sections of each sample were washed again with PBS three times for 5 min; the α-SMA was washed with 3,3′-diaminobenzidine (DAB) and washed with water for 5 min; and the nuclei were stained by immersion in Gill’s hematoxylin solution (Merck Millipore, Billerica, MA, USA) for 6 s.

The slides were dehydrated twice with 95% and 100% ethanol (5 min each) and washed thrice with xylene. Finally, the samples were sealed using Mount Quick Sealant (Takara Bio, Shiga, Japan). The slides were photographed using an integrated stereomicroscope (BZ-X800; KEYENCE, Osaka, Japan). Using EVG staining, the area of elastic fibers was counted using the gridline method; a 100 × 100 µm square was divided into three grid segments at the center of the wound and the average block of the area was measured. The number of α-SMA-positive cells was counted using the gridline method, as previously reported [10]. The average number of α-SMA-positive cells was measured by using the gridline method at five points on the wound.

## 3. Results

### 3.1. Fractional Laser for Scars

#### 3.1.1. Transplanted Cells Took Well after Ablative CO_2_ Laser Treatment

Immediately after the laser irradiation, the scar on the back contracted and became crusted. The surface was crusty and rough, with a bumpy, white, and scaly appearance (Figure 2b). Four weeks after the procedure, the scar widened slightly and became soft and flat (Figure 2c). Additionally, soft hair had grown in some areas.

The fluorescence microscope image of the irradiated scar after 4 weeks showed no fluorescence in the control group and fluorescence in the FB group (Figure 3a–d).

Immunofluorescence staining revealed scattered GFP-positive cells under the skin of the FB group (Figure 3e).

#### 3.1.2. MT-Stained and MMTSS

MT staining showed that the epidermis was thick and disorganized in the control group and mildly thick in the FB group. In the SS group, the epidermis was thick and keratinized, whereas in the LS group the epidermis was thin and near normal. Inflammatory cell infiltration was observed in the dermis in the control group, mild inflammatory cell infiltration in the FB and SS groups, and almost no inflammatory cell infiltration in the LS group. The collagen fibers were pale blue and not dense in the control group; in the FB group, they were blue and arranged in a uniform direction; in the SS group, they were deep blue and dense; and in the LS group, they were blue and formed weaves. Representative MT-stained histological images for each group are shown (Figure 4).

MMTSS was used to calculate the epidermal conditions. Orientation, density, and maturity of collagen fibers in the dermis were evaluated using a score range of 0–14 (2 + (4 × 3)). A score of 0 was considered near normal. Three investigators were blinded to the evaluations (Figure 5).

The results of the evaluation of each of the six mice using the MMTSS are shown in (Table 2). The evaluation averages from the scoring of the three observers are listed in the table; further comparison of the total average values showed that the LS group was closest to normal.

#### 3.1.3. EVG Stained

Representative EVG staining was used to analyze the histological images of normal skin. For each sample, the number of grid blocks with elastic fibers was measured by dividing a 100 × 100 µm square into 100 equal sections with the grid at the center of the tissue section wound (Figure 6).

In the EVG staining for each group, elastic fibers were not observed in the control group or the FB group; in the SS group, the elastic fibers were darkly stained blackish purple. In the LS group, the elastic fibers were wavy, with only a little blackish purple (Figure 7).

The average area of elastic fibers in the samples taken from each of the six mice in each group was measured. T-test results showed that the LS and normal skin groups had similar areas of elastic fibers (Figure 8). Elastic fibers were significantly lower in the control and FB groups than those in the LS group, higher in the SS group than those in the LS group, and significantly lower in the FB group than those in the LS group.

#### 3.1.4. Immunohistochemical Staining of α-SMA

Representative α-SMA immunohistochemical staining of normal skin was used to analyze immunohistochemical images of normal skin. For each sample, the average number of α-SMA-positive cells was measured by dividing a 100 × 100 µm square by the grid at five points of the wound (Figure 9).

In the SMA immunohistochemical staining for each group, the myofibroblasts were observed in the dermis in the control and FB groups; the myofibroblasts in the SS group stained dark brown; in the LS group, few myofibroblasts were observed (Figure 10).

The average number of myofibroblasts in the samples taken from each of the six mice in each group was measured. The number of α-SMA-positive cells in the LS group was similar to that in normal skin (Figure 11). A significant increase in α-SMA-positive cells was observed in the control, FB, and SS groups compared with that in the LS group. Although there was no significant difference, the SS group showed a marked increase in the number of α-SMA-positive cells compared with that in the control and FB groups.

## 4. Discussion

Fractional laser treatment of various types of scars, including hypertrophic and atrophic scars, alters dermal collagen. This dermal collagen alteration results in the restructuring and improvement of the appearance, texture, and flexibility of the scar [11]. Both non-ablative and ablative fractional lasers are available and have been used to treat scars. The non-ablative type is considered safer but has limited efficacy because it acts on dermal collagen through coagulation. The ablative type forms small pores and acts on the dermal collagen [12]. Therefore, there are concerns about post-inflammatory hyperpigmentation; however, it has been reported to be more effective and to have a high degree of patient satisfaction. Traditionally, fibroblast injection therapy has been used in the aesthetic field to improve skin texture [13,14]. Cell transplantation was performed by injecting a cell suspension into the dermis using a syringe. The injection of autologous fibroblasts restored the elasticity of the dermis. This is thought to be the result of an increase in dermal collagen due to the direct effect of the injected fibroblasts and paracrine effect [15,16]. The collagen produced here is thought to be of type 3 rather than type 1. The transplanted fibroblasts also showed the potential for cell division, albeit a limited number of times, which is consistent with the view that fibroblasts from the original skin also proliferate. It has been reported that transplanted cells can be maintained for at least 4–8 weeks [3].

Autologous fibroblast injection therapy was approved by the Federal Drug Administration for the treatment of laughter lines in 2011 [17].

However, injecting many cells locally may cause induration owing to excessive fibrosis, and cell necrosis occurs in some cases. To address these problems, we have developed a new fractional laser cell transplantation method. Fractional lasers can create numerous invisible uniform pores down to the dermis, allowing for more uniform cell transplantation. Percutaneous cell transplantation using a fractional laser was reported in 2014 by Badiavas et al. [18]. This study reports the mechanism by which transdermally introduced cells eventually reach the bone marrow. However, there are no reports of animal studies of successful cell transplantation into the skin using fractional lasers. Since then, no reports have combined the fractional laser and cell transplantation methods.

In our study, the combined treatment with fractional laser and cell transplantation was possible. Figure 3 shows that the transplantation was successful with anti-GFP antibodies. Additionally, the FB group appeared to have less gross scarring than the control group.

We also examined whether differences in culture conditions could reduce scarring. Fibroblasts from fetuses are highly undifferentiated and inhibit scarring [1,4,5]. Wounds are fully regenerated in early embryonic mammals owing to the interaction between macrophages and fibroblasts [19]. Therefore, embryonic fibroblasts were transplanted in this study.

Induction by 3D culture is also possible, as is well-known for hair follicle regeneration [20]. In other words, their undifferentiated nature was preserved. When cells become spherical, a state called cell hibernation occurs. Stem cell-specific membrane surface markers appear during this state of cell proliferation. When fibroblast aggregates are induced to differentiate, they exhibit pluripotency and transform into cells that are equivalent to mesenchymal stem cells in vitro. Therefore, a comparison was made with the three-dimensional culture. Interestingly, the results of this study showed that the elastic fibers were excessively increased in the SS group. The formation of elastic fibers by elastin production is not well understood. Elastin fibers are difficult to reconstruct in vivo [21,22]. Although some reports have shown histological increases in elastin, the increases were mild and often absent. To analyze cell aggregates in two-dimensional cultures and non-adherent culture dishes, it was necessary to match the two cell states as closely as possible. We chose to match the cell counts of the two-dimensional cultures to those of whole-body skin, but we did not confirm whether this approach was correct. Further studies are necessary to clarify why elastic fiber overexpression was observed in the present study.

Notably, after fractional laser irradiation, the tissue sections from the long-term sphere culture group were most similar to the tissue sections of normal skin in all the experiments on the Mouse Masson Trichrome Scar Scale, the elastic fibers, and the myofibroblasts. A sphere is important for maintaining the undifferentiated nature of cells [7,8] and is used for hair regeneration [6]. By creating small holes with a fractional laser, the sphere can be transplanted uniformly into the dermis like a seeding, allowing it to survive. In the future, it will be important to resolve the mechanism by which spherical fibroblasts reduce scarring. Although we counted the number of α-smooth muscle actin-positive cells, we did not use the CD34 assay to test fibroblast maturation. A limitation of our study was that we did not perform immunofluorescence staining for α-smooth muscle actin. This was because immunohistochemical staining with 3,3′-diaminobenzidine obscures shading and limits counting. Our results suggest that combining fibroblast spheres with ablative fractional laser therapy is an effective approach to treating new scars.

## 5. Conclusions

We successfully developed a cell transplantation method using an ablative fractional laser. In addition, we confirmed the differences in the degree of scar reduction depending on cell type. The degree of scarring was significantly reduced by transplanting sphere-forming fetal fibroblasts. However, we did not determine the maturity of the fibroblasts through assays such as the CD34 assay. This issue should be addressed in future studies. It is also necessary to elucidate the reason for the increase in elastic fibers in all the skin sphere groups.

## Figures and Tables

**Figure 1 biomedicines-11-00347-f001:**
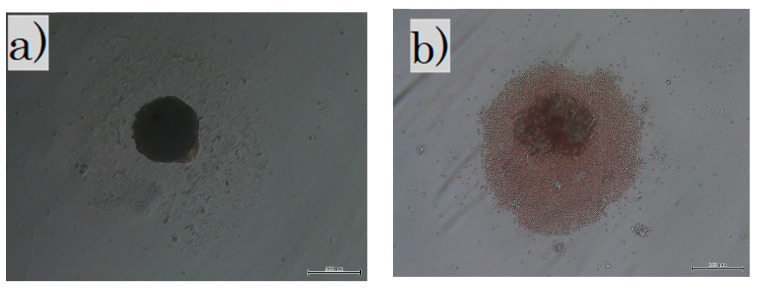
Cells in culture: (**a**) in the LS group, fibroblasts were cultured in non-adherent wells for 3 weeks; (**b**) in the SS group, all skins were cultured in non-adherent wells for 3 days. The dark shadow in the center is a sphere of cells and debris is seen around it. LS, long-term sphere culture; SS, short-term sphere-forming culture (scale bar 200 µm).

**Figure 2 biomedicines-11-00347-f002:**
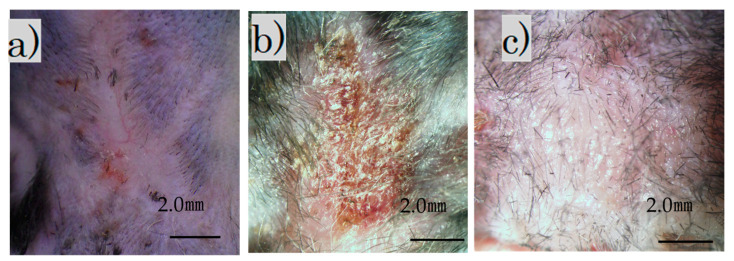
Images of treated and untreated scars: (**a**) mature scar at 2 months; (**b**) scar immediately after irradiation; (**c**) irradiation scar after 4 weeks (scale bar 2 mm).

**Figure 3 biomedicines-11-00347-f003:**
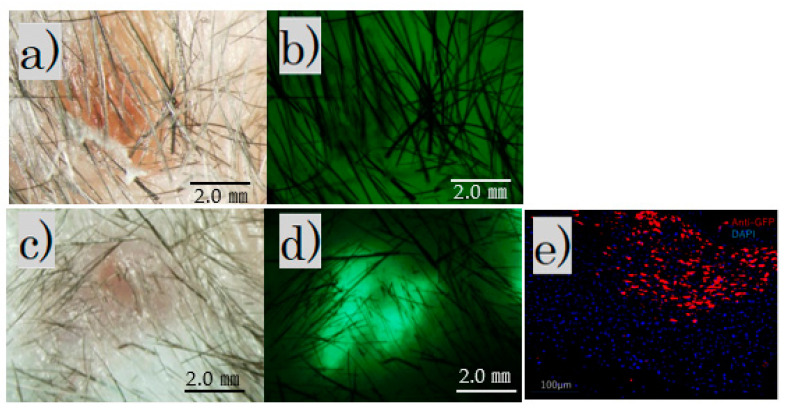
After irradiation for 4 weeks. Fluorescence microscopy: (**a**) control group, bright field; (**b**) control group, no fluorescence observed; (**c**) FB group, bright field; (**d**) FB group, fluorescence observed (scale bar 2 mm); (**e**) FB group paraffin section with immunofluorescence stained by an anti-GFP antibody.

**Figure 4 biomedicines-11-00347-f004:**
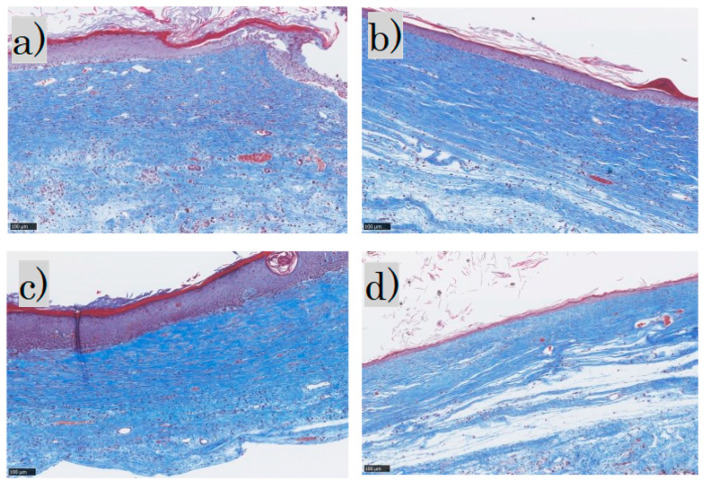
Masson trichrome staining: (**a**) control group; (**b**) FB group; (**c**) SS group; (**d**) LS group. FB, fibroblasts; SS, short-term sphere-forming culture; LS, long-term sphere culture (scale bar 100 µm). Collagenized tissue is stained in blue while other tissues are stained red.

**Figure 5 biomedicines-11-00347-f005:**
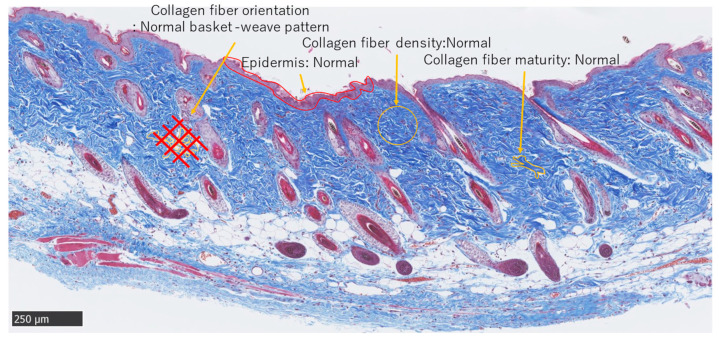
Masson trichrome staining. Normal adult skin (Scale bar 250 µm).

**Figure 6 biomedicines-11-00347-f006:**
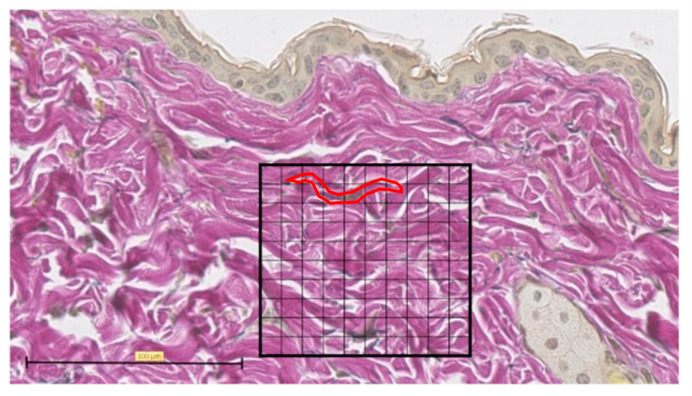
Elastica van Gieson (EVG) staining. Tissue section of normal skin, divided into 100 equal sections of 100 × 100 µm squares (scale bar 100 µm). One typical purple-stained elastic fiber is circled in red and counted as 9 blocks.

**Figure 7 biomedicines-11-00347-f007:**
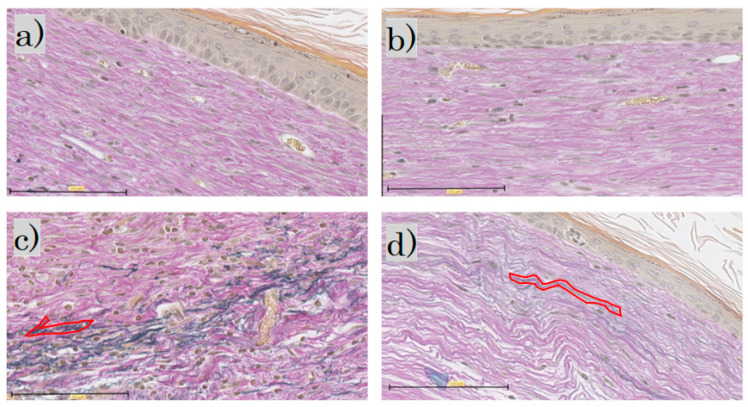
Elastica van Gieson staining: (**a**) control group; (**b**) FB group; (**c**) SS group; (**d**) LS group (scale bar 100 µm). FB, fibroblasts; SS, short-term sphere-forming culture; LS, long-term sphere culture. One typical purple-stained elastic fiber is circled in red.

**Figure 8 biomedicines-11-00347-f008:**
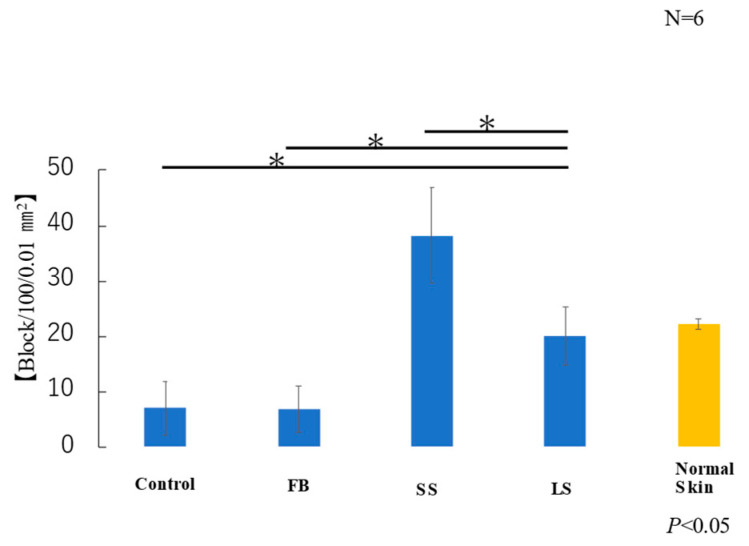
Area of elastic fibers (* *p* < 0.05).

**Figure 9 biomedicines-11-00347-f009:**
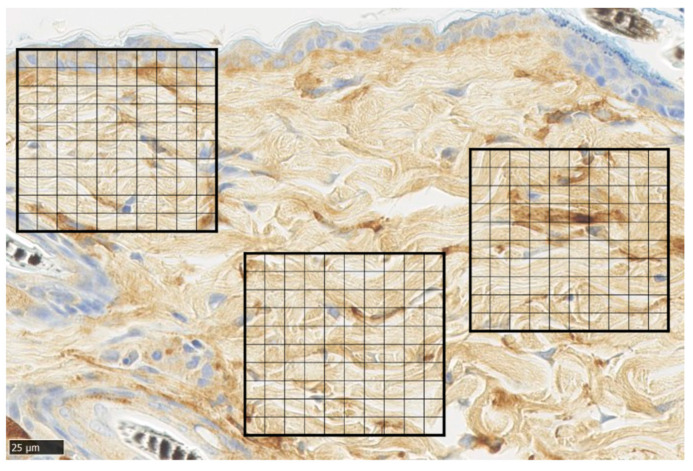
α-smooth muscle actin (α-SMA) staining. Tissue section of normal skin, divided into 100 equal sections of 100 × 100 µm squares (scale bar 25 µm).

**Figure 10 biomedicines-11-00347-f010:**
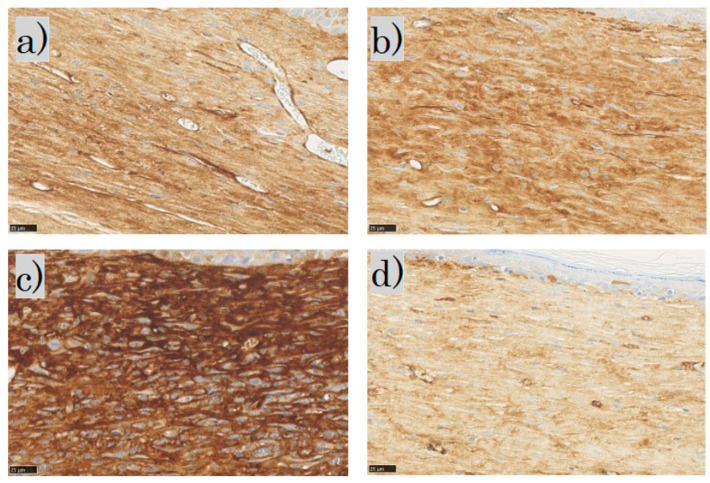
α-smooth muscle actin (α-SMA) staining: (**a**) control group; (**b**) FB group; (**c**) SS group; (**d**) LS group (scale bar 25 µm). FB, fibroblasts; SS, short-term sphere-forming culture; LS, long-term sphere culture.

**Figure 11 biomedicines-11-00347-f011:**
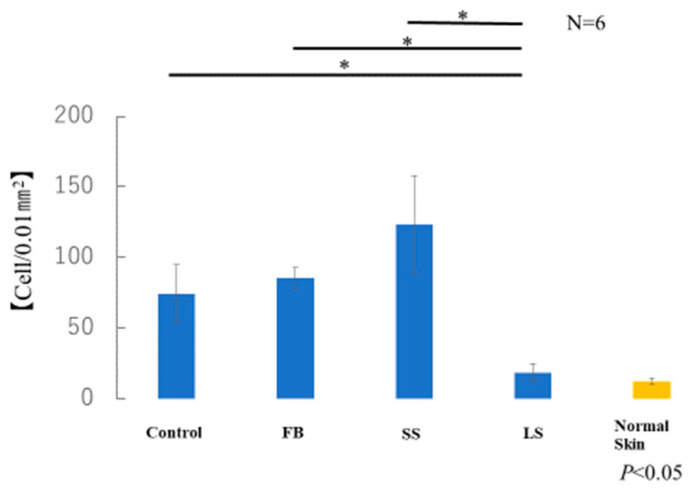
Number of α-smooth muscle actin (α-SMA)-positive myofibroblasts (* *p* < 0.05).

**Table 1 biomedicines-11-00347-t001:** Mouse Masson Trichrome Scar Scale (MMTSS).

**Epidermis**	**B. Collagen fiber density**
Collagen fiber orientation	0 = normal fiber bundle density
0 = Normal	1 ≤ 25% abnormal
1 = Some restoration of rete ridges	2 = 26–50% abnormal
2 = No restoration of rete ridges	3 = 51–75% abnormal
Dermis	4 = 76–100% abnormal
**A. Collagen fiber orientation**	**C. Collagen fiber maturity**
0 = Normal basket-weave pattern	0 = normal fiber maturity
1 ≤ 25% abnormal	1 ≤ 25% abnormal
2 = 26–50% abnormal	2 = 26–50% abnormal
3 = 51–75% abnormal	3 = 51–75% abnormal
4 = 76–100% abnormal	4 = 76–100% abnormal
	Total score range: 0–14

**Table 2 biomedicines-11-00347-t002:** Mouse Masson Trichrome Scar Scale (MMTSS). Average of three researchers’ ratings.

Mouse No.	No. 1	No. 2	No. 3	No. 4	No. 5	No. 6	Total Average
Control	5	8.3	2.3	4.6	10	3.6	5.6
FB	6.3	5	6.3	3.3	4.6	4.3	5
SS	4	3	10.6	2.6	2.3	7.3	5
LS	3.6	2.3	3	5.6	2.3	4	3.5

Abbreviations: FB, fibroblasts; SS, short-term sphere-forming culture; LS, long-term sphere culture.

## Data Availability

Not applicable.

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
