# Peer review of "Fetal Fibroblast Transplantation via Ablative Fractional Laser Irradiation Reduces Scarring"

_biomedicines, 2023, doi:10.3390/biomedicines11020347_

Round 1
Reviewer 1 Report
I have received an interesting paper, by Sakai et al., entitled “Fetal Fibroblasts Transplantation via Ablative Fractional Laser Irradiation Reduce Scarring”. Here are my minor and major comments regarding this article:
Abstract section
- Line 9 – “Scars treatments include fractional laser treatment and cell transplantation” – not so accurate. How about surgery, skin needling, dermal fillers, and so on? – please revise this statement
- Lines 9-10 – “Fractional laser treatment alone only effectively reduces scarring to the point of blurring” – please rewrite this sentence
- Abbreviation should be explained at first appearance in the text (e.g. CO2, GFP)
Introduction section
- This section is scarce and should be revised. More information about scarring pathogenesis and skin regeneration may be considered
- Lines 27-30 – “surgical procedures to reduce….”, “other treatments to reduce…”, “..only effectively reduces..” – please try to avoid repeating words
- Line 33 – please give more data about this “sphere formation”
- Line 35 – try to make a proper context for this information
Material and methods
- Line 41 – how was the skin defect created? Describe the technique because the action mechanism has an important role in scarring formation
- Line 41-42 – provide the features used to consider the “wound healed”
- Line 42 – provide a reason for choosing a cut-off of 2 months after injury
- Line 50 – put a comma after INC.
- Line 51 – explain HEPES
- Lines 53-56 – consider using a diagram to explain the group distribution and better explain the groups. Were there 4 groups?
- Lines 46-47 and lines 59-60 – not sure why repeating this with other words
- Line 57 – “for 3 d…” - explain
- Line 60 – how were the tissue samples collected?
- Line 75 – explain αSMA and H2O2
- As a general conclusion, your study design should be better explained considering that this is an experimental study and the readers should be able to understand the exact methods, with details, of your work, to be able to validate or invalidate your ideas for future research
- Further, considering the nature of the study, a statement about the proper animal care should be presented here
Results section
- Line 94 – “the scar, and the scar contracted became rough like a crust” – please try to better explain this
- Figure 1 should have 2 subsections, maybe A and B, but noted on the picture for a better understanding
- Line 101 – which group is called “the cell transfection group”
- Line 105 – the same comment for “GFP fibroblast transduction group”
- Line 107 – what is DAPI?
Discussion section
- this section should be revised to provide more information. It is not enough to repeat the results of your own study, but to compare it with previous works to increase its value. Even though there are fewer studied about fetal fibroblasts' transplantation, there is current research about fibroblast transplantation and ablative fractional laser use in scar management
Conclusion section
- the conclusion should emphasize somehow that this is an experimental study and the results, even if showed better outcome for the scarring formation, should be further studied, and compared.
Reviewer 2 Report
The article by Sakai, H. et al. about fetal fibroblasts transplantation to reduce scars seems well organized and exposes a relevant message. Authors performed various experiments to demonstrate that a particular combination of treatments is experimentally better to reduce scarring.
The manuscript has several gramatical errors, some of them reported as minor issues. It would probably benefit from a professional review to correct them. The style is not particularly bad, but I miss a lot of text to correctly explain the experiments.
In my opinion the experiment with SMA (actin?) using simple immunohistochemistry has limited relevance to assess the number of fibroblasts; I think the correct technique for this purpose should be SMA immunofluorescence, counterstained with DAPI to be really demonstrative (that was already employed in some experiments). I thus, suggest to perform this additional experiment. However, I agree with the interpretation. Finally, I miss using a CD34 assay to investigate the maturity of the fibroblasts.
There are some additional major issues:
- I miss an explanation of the employed histochemical methods in materials & methods (Masson and Van Gieson).
- Images and Figures describing the results should be included after their explanation in the text. You usually read the results and, thereafter, you look the images to demonstrate those results.
There are many minor issues as well:
- Abstract, general. Please, don’t repeat exactly the same sentences in abstract and the main text. For example there is nearly the same sentence to begin the abstract and the introduction. You can maintain the idea, but do creative work with the construction of the sentences.
- Materials, page 1, lines 41-42. I suppose that the wounds involved in the experiment were left untreated during the 2 first months, and then the laser was applied. If true, I think this should be added to the text.
- Materials, page 2, lines 46-47. You should be more precise with the sentence “fibroblasts were applied”… Where in the irradiated area? In the middle? In all the area? How were applied those fibroblasts? Covering the skin? With an invasive method? You talk about this later, but you should be more precise here. You may also remove or change the sense of this particular sentence.
- Materials, page 2, line 75. What’s alphaSMA? Smooth Muscle Actin? Please define the acronyms.
- Materials, page 3, line 84. I miss a word here: “sections were water”.
- Results, page 3, line 94. You can change this line for something like this: “The scar contracted and became crusted.”
- Results, page 3, line 99. You depict the scar after 4 weeks, but 4 weeks taken from the wounding or from the irradiation procedure? The same for Figure 2.
- Results, page 3, lines 100-105. You should be much more informative and clear with the description of this experiment. Please, include more text for this purpose.
- Results, page 4, lines 106-108. In materials and methods you describe indirect immunohistochemistry, but here you employ non-described immunofluorescence. In addition “They were immunostained with…”.
- Results, page 4, Figure 3. This image is poorly described.
- Results, page 4, line 111. MT was not described as Masson´s Trichrome anywhere before this title.
- Results, page 4, line 119. Please, employ past tense when referring the results of your experiment.
- Results, page 5, Table 2. I can’t understand this table. You have 8 animals, with 4 possible procedures, and three observers. I guess that row No.1 refers to observer 1 in animal 1, No.2 to observer 2 in animal 1, No.3 to observer 3 in animal 1, No.4 to observer 1 in animal 2. But just guessing! The table is quite unclear.
- Results, page 5, line 132. Paraffin blocks? Fibers WERE measured (this issue is widely present). Afterwards, you are describing a grid. But I have no idea about the meaning of the grid, did you employed it for calculating the area of elastic fibers?
- Results, page 5, Figure 7. Another figure poorly described. What does it mean the asterisk? The bars and the lines just above the bars? The lines above the figure?
- Results, page 5, line 142. SMA immunohistochemistry.
- Results, page 6, Figure 9. The same issues mentioned in Figure 7.
- Discussion, page 7, lines 165-175. I miss a lot of bibliography here to support the performed statements.
- Discussion, page 7, line 162. I hope you are referring to bone marrow precursor cells.
- Discussion, page 7, lines 163-164. You state that “transplanted” cells seems to be introduced in the skin, but the cited reference mentions “percutaneous transplantation”. This should be clarified. I think you should distinguish this method from microneedling, as long as the final result seems pretty similar.
- Discussion, page 7, last paragraph. Please, do not include the acronyms here, as this final paragraph seems the actual conclusion of the study and they difficult a quick reading.
Reviewer 3 Report
Comments:
Authors combined the ablative fractional laser irradiation with fetal fibroblast transplantation to treat scars and reported promising findings, but manuscript requires attention on following comments:
1) Materials and Methods, Section 2.2. ‘Fibroblasts culture and transplantation’, the section looks fragmented, not coherent. Tissue collected, maintained by explant method (not described properly, is that method described elsewhere, if so cite the reference or describe briefly). Then, cells after 3 passage were used. How cells were isolated?, later it was mentioned that cells were isolated with collagenase. Which collagenase was used e.g. Type I, II, III etc.? How the contaminating keratinocytes or other skin cells were removed ? What is the composition of a non-adherent culture media and how spheres (long-term sphere culture group (LS) vs short-term sphere culture (SS)) were made, is there any difference in methodology of two types of spheres? What quantity of cells or spheres were used in the transplantation? “protected with film” what kind of film, is it prepared by authors or commercially available film was used, describe in detail.
All these information is critical to reproduce the experiments and methodology should be written sequentially and coherently.
2) Figure 1, requires some improvement, authors should ideally remove the back hair (2-3 cm) around the scar, so that changes can be easily observed and interpreted. Authors should also share the representative picture of mouse back (after 4 weeks) where no treatment was applied to compare the efficacy of treatment.
3) Page 4, line 102, ‘cell transfection group’ ? transfection or transplanted?, similarly word ‘transduction was used’? clarify the reason of using different terminology?
4) The figure order and legends were not appropriately placed in the manuscript, creating confusion
5) English/Grammar requires attention in the manuscript. Authors used both past and present tense?
6) Figure 4: Use arrows to show the differences between control and treated group (e.g. fibrosis)? If possible include Masson’ s Trichome staining of normal skin. It will be interesting to see the MMTSS scoring of normal skin (no scar, untreated). Include representative photograph of normal skin in figures # 4, 5 and 8.
7) Methodology should be clearly defined, how the images of GFP in intact skin were taken?
8) Line #194 mentioned normal skin, please define the controls properly. It is confusing whether control refers to ‘Normal skin’ or ‘untreated scar skin’.?
9) Authors reported increase in Elastin fibers SS group, author should discuss the significance of Elastin fibers in normal skin using appropriate references whether it confers benefit in skin homeostasis.
Reviewer 4 Report
This study aimed to devise a treatment combining ablative CO2 fractional laser treatment with cell transplantation to investigate how to treat scars.
Eight-week-old male 13 C57Bl/6 mice were used to create a full-layer skin defect in the back skin and make scars. The scar was irradiated with a CO2 fractional laser. After that, cells were transplanted on the scar surface and sealed with a film agent.
Transplanted cells were GFP-positive murine fetal fibroblasts (FB), fetal 16 fibroblasts with long-term sphere-forming culture (LS), and fetal skin with short-term sphere-form-17 ing culture (SS). After transplantation, the authors found that GFP-positive cells were scattered in the dermal papillary layer and subcutis in all groups. LS significantly reduced the degree of scars, which were 19 closest to normal skin.
The authors conclude that combining ablative fractional laser irradiation and fetal fibroblast transplantation allowed us to approach new methods for scar treatment.
My questions are the followings:
a) Please make clear about the introduction. The part of this paper is essential to follow the procedures of the manuscript
b) Where is Fig. 2?
c) Why in the material and methods you don't make mention about the use of cryostatic sections. This mean that you used frozen biopsies? Why is not added to this paragraph?
d) Why in the paragraph related to immunohistochemical procedures you miss the method for anti GFP staining?
e) Why for figures 4, 5, 8 you have not added the letters inside the pictures?
f) Please comment the results of Fig. 3
g) Please comment the results of Fig. 4
h) Please comment the results of Fig. 5
i) 3.1.2. MT-stained and MMTSS Representative MT-stained histological images of each group are presented (Figure 4).
Why didn't you describe the figure placed between figure 3 and figure 4? Please describe the result, place some letters and an accurate legend.
Round 2
Reviewer 1 Report
Congratulations on the work you have done! I think the explanations, comments and new information have increased the value of this study and made it easier to understand.
I have only three minor comments:
- Figures 2, 3 - if the line at the bottom of the image is a scale, it should be graded
- Figures 6, 7 – the gradation is undecipherable
- some punctuation marks need to be corrected
Author Response
Reviewer 1
Comments and Suggestions for Authors
Congratulations on the work you have done! I think the explanations, comments and new information have increased the value of this study and made it easier to understand.
I have only three minor comments:
- Figures 2, 3 - if the line at the bottom of the image is a scale, it should be graded
- Figures 6, 7 – the gradation is undecipherable
- some punctuation marks need to be corrected
Submission Date
30 November 2022
Date of this review
05 Jan 2023 10:52:53
Thank you very much for your kind peer review.
- Figures 2, 3 - if the line at the bottom of the image is a scale, it should be graded
We have added scale bar units to these lines, which are shown.
- Figures 6, 7 – the gradation is undecipherable
We have added the following text.
Figure 6: One typical purple-stained elastic fiber is circled in red and counted as 9 blocks.
Figure 7: One typical purple-stained elastic fiber is circled in red.
- some punctuation marks need to be corrected
We have reviewed the manuscript again to correct these errors.
Reviewer 2 Report
The article by Sakai, H. et al. about fetal fibroblasts transplantation to reduce scars was notably improved. After the improvements, particularly in table 2, I can now see that every mouse experienced all the treatments of the experiment.
The manuscript still has various language errors, some of them reported as minor issues. It would definitely benefit from a professional review to correct them.
I still have various major issues with the text:
- There is no fluorescence in controls, but I do see fluorescence in the FB group. Actually, in Figure 3 I see lesser scarring in FB group and also fluorescence, but I miss a statement about it and a posterior discussion.
- The absence of SMA immunofluorescence should be clearly noted as a limitation (and include the word “limitation” or “limited”), as it actually limits the counting. You have to keep in mind that the “intensity” of staining may be influenced by several laboratory technical aspects. Fluorescence is quite more homogeneous and displays much better the nuclei for counting. These ideas should be included after the limitation statement.
- I still miss a good explanation of the employed histochemical methods in materials & methods (Masson and Van Gieson). Probably a separate section in materials and methods before MMTSS, describing both techniques and their results, would be helpful.
- Results, lines 191-193. I don´t know what kind of Van Gieson stain did you perform, as long as it was not defined in materials. But Van Gieson trichrome was designed to stain collagen in red. It may be combined with the monochrome Verhoeff stain, which stains black the elastic fibers. Please revise the referred text, figure 6 and the legend. I can hardly see any elastic fiber in the proposed images (may be in the SS group, it´s true), but there are lots of collagen fibers (which is normal). You should explain the results, and consider a broader discussion. I think an insert in the figure with higher magnification would be fine to be demonstrative of the elastic fibers (that have an “S” shape).
There are many minor issues as well:
- Abstract, lines 8-9. This first statement is already well known. It is not attractive as the first sentence in an abstract. You can remove it and the manuscript will be better.
- Abstract, line 10. “Effectively reduce”; I think you are generalizing here and not referring to a particular experiment (which is not mentioned).
- Abstract, line 11. I guess you are referring here to mature fibroblasts.
- Introduction, line 26. Another deceptive first sentence, it is clear you have been studying skin regeneration as long as you make various self-citations. You don’t need to state your CV, just the experiment. This sentence is useless.
- Introduction, line 38. “Transcription factors were markedly elevated”… All the transcription factors? Some key factors? Please specify.
- Introduction, lines 36-38. You should merge the sentence in lines 40-41 with the mentioned one, before referring the “preliminary experiments”, as it is a bit confusing to read.
- Materials, line 64. “The scar site with fractional laser cultured cells”. Here, the sentence seems more logical if you remove “the scar site with fractional laser”.
- Materials, line 68. Mice were not “euthanized”. They were “sacrified”… Please, check the meaning of “euthanasia” in a dictionary (you can use Oxford).
- Materials, figure 1. I can´t find the purpose of these images. Are they demonstrative of the cellular growth? To my eyes figure 1b is greater than figure 1a, which is not explained… Honestly, I recommend to remove this figure.
- Materials, line 120. What kind of “professional”? This is vague information, you should specify… I suggest to remove this word: “evaluated using light microscopy”.
- Materials, line 140. If you choose not to directly explain the gridline method, you are expected to include at least one reference.
- Results, line 144. “And became crusted”
- Results, line 147. “soft hair grew”.
- Figure 2. You should include similarly sized photos, please pay attention to this point in all the figures. In addition, the order is important, you should begin with the scar after irradiation, continue after one month and finish with the scar at 2 months.
- Results, lines 153-155. This paragraph belongs to materials section, not to results.
- Results, lines 161-162. You already defined MT as Masson´s trichrome, so you can include the acronym if you want.
- Figure 4. As long as we are dealing with a general scientific journal, it may be useful to specify that collagenized tissue is stained in blue while other tissues stain red with Masson.
- Results, line 180. I think that reference to Figure 5 should be included here, and not in line 182.
- Results, point 3.1.3. This point seems disordered. The first paragraph should be the one beginning with “representative…”, followed by figure 7. The second paragraph should be the one beginning with “in the EVG staining…”, followed by figure 6. The same logic should be applied to point 3.1.4.
- Results, line 215. “With SMA immunohistochemical staining…”
- Figures 7 and 9. These are the same grid. The scale bars seem good, but the grids seem clearly out of scale. You define a grid composed by 100 equal sections of 100 x 100 square micrometer squares. These images are clearly out of the scale of the grids!
- Discussion, lines 246-251 (at least). I STILL miss a lot of bibliography here to support the performed statements.
- Discussion, lines 263-264. “However, the cells that remain in the skin have not been reported” seems not logical, may be some text is missing.
- Discussion, lines 272. In the same line is repeated “In other words”.
- Discussion, lines 294-296. This newly added text is already added in conclusions. You can remove it.
Reviewer 4 Report
The authors have answered correctly to my questions
Author Response
Thank you for the peer review.
It was very informative.